# Thwarting Instant Messaging Phishing Attacks: The Role of Self-Efficacy and the Mediating Effect of Attitude towards Online Sharing of Personal Information

**DOI:** 10.3390/ijerph20043514

**Published:** 2023-02-16

**Authors:** Yi Yong Lee, Chin Lay Gan, Tze Wei Liew

**Affiliations:** Faculty of Business, Multimedia University, Melaka 75450, Malaysia

**Keywords:** self-efficacy, attitude, phishing susceptibility, anti-phishing knowledge, protection motivation theory

## Abstract

Context: The cause of cybercrime phishing threats in Malaysia is a lack of knowledge and awareness of phishing. Objective: The effects of self-efficacy (the ability to gain anti-phishing knowledge) and protection motivation (attitude toward sharing personal information online) on the risk of instant messaging phishing attacks (phishing susceptibility) are investigated in this study. The protection motivation theory (PMT) was tested in the context of attitudes toward sharing personal information online with a view to improving interventions to reduce the risk of phishing victimisation. Methods: Data were collected using non-probability purposive sampling. An online survey of 328 Malaysian active instant messaging users was collected and analysed in SmartPLS version 4.0.8.6 using partial least squares structural equation modelling. Results: The results showed that a person’s cognitive factor (either high or low self-efficacy) affected their chance of being a victim of instant message phishing. A higher level of self-efficacy and a negative attitude towards sharing personal information online were significant predictors of phishing susceptibility. A negative attitude towards sharing personal information online mediated the relationship between high levels of self-efficacy and phishing susceptibility. A higher level of self-efficacy led to the formation of negative attitudes among internet users. Attitudes toward the sharing of personal information online are critical because they allow phishing attempts to exist and succeed. Conclusions: The findings give government agencies more information on how to organise anti-phishing campaigns and awareness programmes; awareness and education can improve one’s ability to acquire anti-phishing knowledge (self-efficacy).

## 1. Introduction

Internet communication technologies have changed the nature of para-social interaction from passive to an approximation of concrete or real social interaction [1]. Internet communication technologies (i.e., instant messaging applications) are the most popular social media applications among Malaysian citizens [2,3]. Sending daily messages, group messages, and other forms of communication via the internet has become a user’s daily routine in order to facilitate social activities and relations [4,5]. Phishing, such as click baits (links or text embedded in messages or emails to entice users to view and read, with the intent of deceiving internet users), is an ongoing issue [6,7]. Clicking on attached links in instant messages without verifying the source could unintentionally lead internet users into a phishing trap [6,8,9]. According to the International Criminal Police Organization (also known as Interpol), phishing attacks in ASEAN countries show no signs of abating or slowing down [10]. Kaspersky, a cybersecurity and anti-virus software company, has successfully blocked over 1.6 million phishing attempts. This has kept internet users safe from phishing attacks in the modern era [10]. Kaspersky’s anti-phishing systems stopped 12,127,692 malicious links in South-East Asia from January to June 2022, an increase of one million over the 11,260,643 malicious links that were discovered during the same time period in the previous year. Phishing attacks in South-East Asia (i.e., Malaysia) outnumbered those in the previous year [11]. Kaspersky also identified and thwarted 91,895 similar assaults in the first half of 2022 that were made against Malaysia’s 27,458 banks [12]. In Malaysia, phishing threats are on the rise [13], and the majority of phishing victims were vulnerable to instant messaging phishing attacks [14,15,16,17,18]. In Malaysia, phishing attempts increased in the first six months of 2020, accounting for 749, 915 cases, compared to the first half of 2019 [10]. Phishing attacks on social media platforms increased by 20% in the second quarter of 2020, compared to the first three months of the year [10]. The phishing attacks were primarily motivated by attacks against WhatsApp and Facebook [10]. Therefore, the purpose of this study is to investigate the factors that influence the risk of phishing victimisation, specifically the risk of instant messaging phishing victimisation.

According to Das et al.’s [19] meta-analysis, while it is critical to provide technological solutions, such as warning indicators and browser extensions, or designing games-based solutions to thwart spear-phishing attacks [20], identifying specific human traits that a phisher can use to successfully exploit the user is critical for detection, prevention, and mitigation techniques. Psychology research on victimised users should focus on their mental models and the characteristics that make them vulnerable to such attacks [19,21]. In Malaysia, there is a growing corpus of literature and study on the consequences and challenges of phishing attacks [22,23,24]. The most significant consequence will be financial losses [13,24]. Phishing threats can cost internet users money as well as the costs of organisation support [25]. Specifically, phishing threats have exposed vulnerable victims to risks or dangers, as well as consequences [26]. Aside from monetary losses, customers will lose trust in a company if they believe legitimate messages are phishing messages [25]. Prior empirical research indicated and measured the level of cybercrime awareness among Malaysians [27]. According to the findings, respondents had little knowledge of phishing scams and were unaware of them [27]. The cause of cybercrime phishing threats in Malaysia is a lack of knowledge and awareness of phishing [21]. When an internet user has a lower level of cybercrime risk awareness, knowledge, or skills, they will be less cautious, resulting in fraud victimisation and monetary losses [28]. Internet users must therefore exercise caution when using the internet because they are essential to establishing online security [29,30].

Self-efficacy, which refers to people’s perceptions of what they can accomplish with their abilities [14], is linked to knowledge [31]. For example, when internet users are more confident in taking the necessary precautions to avoid phishing attempts (self-efficacy), they are aware and knowledgeable of the risks involved, resulting in the avoidance of phishing attacks [31]. For example, a high level of ability to acquire anti-phishing knowledge (self-efficacy) may reduce the risk of phishing victimisation [32]. According to Hameed et al.‘s [33] meta-analysis, researchers confirmed that self-efficacy is a key predictor in the context of information system security, particularly in mitigating the risk of information system security. In contrast, researchers discovered a significant and positive relationship between self-efficacy and susceptibility to phishing [34,35,36]. Internet users who have a high level of self-efficacy are more vulnerable to phishing attacks. People with a high level of self-efficacy are less likely to recognise a security attack [36] because they are overconfident in their ability to detect online fraud or phishing [37]. As a result, it begs the question of whether self-efficacy is related to one’s susceptibility to phishing victimisation.

Additionally, cybercrime incidents can be reduced if internet users are knowledgeable about cybersecurity [38]. Precautionary online behaviour, for example, is also necessary for achieving online security [29,30]. A lower level of self-efficacy has been linked to lower levels of protection motivation and behaviour [39], for instance, people are more likely to share personal information online, exposing themselves to phishers [32]. According to Jansen and van Schaik’s [30] research, there is a scarcity of research on behavioural or attitude change interventions for cybersecurity. Precautionary online behaviour (i.e., a negative attitude toward the sharing of personal information online) may aid in preventing phishers from impersonating or deceiving internet users [32]. Therefore, this paper aims to investigate the relationship between self-efficacy and attitude towards precautionary behaviour.

The protection motivation theory (PMT) served as the foundation for attitude toward protective behaviour, which is more specifically operationalised as attitude towards sharing personal information online [32,40,41]. When determining whether someone will engage in security behaviour, attitude acts as a mediator [42,43]. It was discovered that a negative attitude toward sharing personal information online was a key predictor as well as a mediator in predicting phishing vulnerability in order to encourage security behaviour, particularly against phishing attacks [30]. Nowadays, the internet makes it easier for individuals to communicate and share personal information on social media [6]. However, when it comes to online privacy, internet users are having difficulty protecting their personal information [44]. As cybercrime becomes more prevalent, there is an urgent need to educate people about the risks of excessive information sharing, information control, information visibility, and privacy issues [45]. In today’s rapidly evolving digital environment, user privacy has emerged as a critical issue that must be addressed [46]. A recent study recommended a game-based strategy to teach people about the dangers involved in information sharing on the internet [44]. Notably, the purpose of this research is to investigate the factors that contribute to the risk of phishing victimisation. Identifying the mediator which can reduce or mitigate the risk of phishing victimisation is therefore critical. With a nod to and theoretical justification from PMT, the attitude toward protection behaviour (attitude toward sharing personal information online) was chosen as a predictor as well as a mediator in this study.

The subsequent Section 2 discusses the theoretical foundation, which is then followed by the development of the research model and a discussion of the hypotheses. Section 4 describes the research methods, while Section 5 discusses the results. Section 6 explores the discussions and implications. The study’s limitations are covered in Section 7, and the study’s conclusion is presented in Section 8. 

## 2. Theoretical Background

### Protection Motivation Theory 

Protection Motivation Theory (PMT) was established by Rogers [47]. PMT was developed to illustrate and comprehend individuals’ risk-aversion behaviour in the field of health research [32,42]. PMT has been used to gain better insights into the intention to engage in protective behaviour [30,32,40]. Similarly, Warkentin et al. [48] presume that PMT assists in developing communication strategies that encourage people to take precautionary measures to avoid becoming a victim of cybercrime.

In the PMT, two major cognitive processes have been recognised (i.e., threat appraisal and coping appraisal) to predict attitude towards protection behaviour [32,40]. Threat appraisal was operationalised as perceived vulnerability/ susceptibility and perceived severity [32,49]. Coping appraisal was operationalised as self-efficacy or response efficacy [30,32,49].

A threat appraisal refers to evaluating a specific threat and the risk it entails [32]. For example, threat appraisal was operationalised as perceived severity in order to investigate its relationship with perceived vulnerability or the “likelihood of being victimised by a specific cybercrime threat” [32,50]. Coping appraisal is the process by which an individual evaluates various methods of protection. Personal ability to comply with protection methods (i.e., self-efficacy) and effectiveness of protection methods (i.e., response efficacy) are two examples [32]. 

Perceived severity was defined as the “extent someone believes that the consequences of threats would be harmful, increases the motivation toward protecting oneself against those threats” [32,41,47,51,52]. In other words, the more harmful the individuals perceive the threats, the more they would desire to perform security measures in order to avoid becoming cybercrime victims [32]. According to the definition, perceived severity was used to assess the outcomes or consequences of a threat, particularly phishing attacks [30,32].

Perceived vulnerability or susceptibility, however, was defined as “the risk or likelihood that an internet user will be deceived by a cybercrime attack” [32,53], particularly, phishing threats [54]. Perceived susceptibility was adopted in the current study as the dependent variable to measure one’s risk of phishing victimisation. In other words, perceived susceptibility acted as a dependent variable to assess the perceptions of the respondents towards phishing susceptibility [9,54,55]. These perceptions include the possibility, probability, or risk/likelihood that the respondents think they will become phishing victims [32,54].

From a cybersecurity standpoint, self-efficacy plays a vital role in motivating cybersecurity protection behaviours [30,56]. When an individual possesses a high level of self-efficacy, he or she can detect phishing threats easily and manage to identify the cues, such as content authentication cues and sender verification cues [54,57]. These cues were used by internet users to validate the authenticity of phishing messages before completing any tasks requested by strangers, such as disclosing personal information [54].

In addition, response efficacy was defined as “an individual’s evaluation of the perceived effectiveness of the recommended response” [30]. Response efficacy was used to assess the final outcome of whether a particular safety or security measure could effectively prevent phishing attacks [30]. Since the objective of this study was to identify the factors influencing perceived susceptibility to phishing victimisation, perceived severity and response efficacy were not included as an independent variables. This is due to perceived severity being used to measure the severity of the threat, focusing on the victims’ perceptions of the consequences of being victimised, and not being victimised [30]. 

A growing number of empirical studies have recently substituted “attitude toward protection behaviour” for the original term “protection motivation” [32,40,41]. PMT has recently been adopted in the field of information security [30,58] as a mediator by adopting an attitude toward protection behaviour [32,40]. When PMT is used as a theoretical foundation for interventions, attitude plays an important role in avoiding cybercrime phishing victimisation [30]. The current study adopted the viewpoint of Jansen and van Schaik [30], who examined phishing susceptibility and identified attitude as a potential mediator. Figure 1 shows the current study’s overall research framework.

## 3. Research Model and Hypotheses Development

### 3.1. Self-Efficacy, Attitude, and Phishing Susceptibility

Self-efficacy is defined as “a person’s confidence in taking the precautionary measures, that is, the perceptions of one ability in protecting oneself online” [31]. Internet users have low levels of perceived privacy self-efficacy, implying that they consistently believe they have little control over their personal information [59]. Higher levels of perceived self-efficacy are associated with increased protection motivation and behaviour [39,47]. Indeed, studies have shown that higher levels of self-efficacy led to increased online security measures, such as communicating safely with others online [60]. In the field of information systems, self-efficacy has been extensively researched [54]. In the context of phishing, self-efficacy was identified as an important driver that could reduce the risk of phishing victimisation. For the purpose of this study, self-efficacy was adopted as researchers have indicated that it is a vital antecedent of attitude toward protection behaviour [30,32,40] and phishing susceptibility [6,61] respectively. 

Perceived self-efficacy in knowledge related to phishing attempts significantly and negatively impacts an individual’s likelihood of responding to phishing emails [62]. Previous research has acknowledged self-efficacy as a crucial antecedent to thwarting phishing attacks [63]. This is due to the fact one is often inclined to avoid security threats by adopting online precautionary measures when he or she believes that such security measures can be successfully implemented [63]. Verkijika [63], on the other hand, contends that research on the impact of anti-phishing self-efficacy on mobile phishing avoidance behaviour is limited. Since the present study was conducted to examine phishing victimisation, particularly on instant messaging phishing, anti-phishing self-efficacy was adopted as one of the independent variables to examine whether it is applicable in explaining phishing conducted via instant messaging. 

Recent empirical studies have found that individuals with a high level of self-efficacy have a lower risk of becoming a victim of crime [64]. Numerous studies have found that self-efficacy influences phishing susceptibility. Recent empirical studies have found that individuals with low self-efficacy are more vulnerable to cyber-social engineering victimisation [65,66]. Researchers, on the other hand, discovered a significant and positive relationship between self-efficacy and phishing susceptibility [34,35]. In light of the contradictory findings, the purpose of this study is to investigate the relationship between self-efficacy and phishing susceptibility. The current study, guided by protection motivation theory [47], seeks to determine whether the influence of self-efficacy can reduce the risk of phishing victimisation [30]. As a result, this study hypothesised the following:

**H1.** *A higher level of self-efficacy leads to a lower risk of instant messaging phishing victimisation. (i.e., there will be a negative relationship)*.

It has been demonstrated that self-efficacy influences an individual’s attitude toward protective behaviour [30,51]. In other words, increased self-efficacy leads to a more positive attitude toward protective behaviour, specifically a negative attitude toward online information sharing [32]. This is because when a person believes in his or her own ability to perform a behaviour, he or she is motivated to engage in the protection behaviour [41,67] and thus declines to share personal information online [32]. As a result, the following hypothesis was proposed in this study:

**H2.** *A higher level of self-efficacy leads to a negative attitude towards sharing personal information online. (i.e., there will be a positive relationship)*.

### 3.2. Attitude and Phishing Susceptibility 

Under the guise of ignorance, individuals’ risk attitudes influence their final decision-making [68]. A risk attitude, for example, may influence one’s ability to recognise phishing attacks correctly [68]. However, there are few studies that look at people’s attitudes in the context of cybersecurity research [69]. The attitude of internet users toward cybersecurity issues and cyber deception may influence their vulnerability to cybercrime [70]. Furthermore, people with a high desire to gamble (risk attitude) are more likely to click on phishing messages sent by scammers and fall victim to phishing attacks [71].

Attitudes toward protective behaviour can be critical in raising internet users’ awareness of threats [30]. One’s attitude can provide behavioural advice on how to process phishing messages and mitigate the threat, especially in cybercrime phishing attacks [30,32]. Because the purpose of this study is to investigate the risk of instant messaging phishing victimisation, it is critical to raise individual threat awareness in order to mitigate or lower the risk. Aside from threat knowledge, one should cultivate an attitude; a positive attitude toward protective behaviour (i.e., not sharing personal information) is essential in security behaviour [30]. Individuals will have a positive attitude toward protective behaviour when they perceive their vulnerability to becoming a cybercrime phishing victim [32,47]. 

Furthermore, previous research has shown that attitude is an important factor in predicting burnout and violent victimisation [72]. There are, however, few studies that use attitude as an independent variable to predict phishing victimisation. As a result, the purpose of this study is to use attitude as an exogenous variable to investigate instant messaging phishing victimisation, hence the third hypothesis:

**H3.** *Having a positive attitude towards sharing personal information online leads to a higher risk of instant messaging phishing victimisation. (i.e., there will be a negative relationship)*.

### 3.3. The Mediating Role of Attitude towards Sharing Personal Information Online

The attitude was operationalised in the context of cyber-dependent crime (phishing) as the attitude toward sharing personal information online [30,32,67]. When a person has a positive attitude toward sharing personal information online (revealing information to strangers), he or she is more likely to become a victim of phishing attacks [32]. Nowadays, the internet’s penetration and widespread adoption of social media platforms provide a fertile ground for phishing scams [73,74,75]. Internet users are at high risk of becoming cybercrime victims due to a lack of cybersecurity awareness about online threats and an increased positive attitude toward sharing personal information online [63,73].

A previous study in the cyberspace research context discovered that limiting the visibility of profiles in privacy settings had a significant impact on attitudes toward self-disclosure [76]. There is evidence that an internet user who is concerned about online security management does not necessarily limit self-disclosure in cyberspace, but rather has a negative attitude toward sharing personal information online [77].

Aside from technical safeguards, numerous studies emphasise the importance of attitude in understanding how target victims fall victim to phishing scams [32,78]. Attitude conceptualisation was initiated from the theory of planned behaviour [79] and the theory of protection motivation [32]. Attitude influences risk-taking in relationships, which in turn influences the processes and outcomes [78]. Individual attitudes may have a significant impact on their security perceptions [80].

An individual’s ability to gain anti-phishing knowledge—namely, self-efficacy—influenced his/her risk of phishing victimisation significantly [65]. However, because of a lack of in-depth information, a message receiver with less prior knowledge may be influenced by phishing message cues [81]. As a result, this study aims to investigate the role of attitude in mediating the relationships between self-efficacy and phishing susceptibility (risk of instant messaging phishing victimisation). Thus, the fourth hypothesis is as follows:

**H4.** *Attitude towards sharing personal information online mediates the relationship between self-efficacy and the risk of instant messaging phishing victimisation*.

## 4. Method

### 4.1. Participants

All participants were recruited using social media platforms (i.e., Facebook, WhatsApp). Non-probability purposive sampling was used for data collection. The respondents had to be older than 18 and commonly communicate online via instant messaging applications in order to qualify as respondents. This study received 335 questionnaires in total. Following a thorough examination of the 335 datasets, a total of six datasets were detected with straight lining (3 and 5 s) and were omitted [82]. One response stated that they did not use instant messaging platforms on a regular basis. Hence, seven responses were excluded, leaving a final sample of 328 for the subsequent analyses. This resulted in a 97.9% response rate, far exceeding the 80% statistical power suggested by G*Power, which suggested that a sample size of 68 would be sufficient.

Finally, a valid sample of 328 participants was used, with 151 men (46%) and 177 women (54%). They were between the ages of 18 and 43 (mean = 23.78, standard deviation = 3.99). In terms of years of education, 10.4% had 11 to 14 years (secondary school, diploma holder), while the remaining respondents had 15 to 20 years (bachelor´s degree, master´s degree, PhD). The majority of the respondents were currently residing in the centre of Malaysia (N = 155; 47.3%), followed by the Southern region (N = 78; 23.8%), Northern region (N = 47; 14.3%), East Malaysia (N = 36; 11%), and the East Coast (N = 12; 3.7%). In terms of years of instant messaging use, 30.2% had 1 to 6 years of experience using an instant messaging platform, while 69.8% had more than 6 years of experience using an instant messaging platform. In terms of occupation, 80.2% were students (N = 263), 18.3% were currently employed (N = 60), and 0.9% were unemployed (N = 3). More than half of the respondents (60.4%) did not have a monthly income. This percentage corresponds to the previous percentages for the occupational group, with the majority of the group being students. 

The top three instant messaging platforms for online communication were WhatsApp (N = 292), Facebook Messenger (N = 225), and Telegram (N = 143). A total of 6% of those surveyed said they received phishing messages more than once a week (usually receiving phishing messages). One hundred and fifty-eight people stated that they get phishing emails once or twice a month (sometimes receiving suspicious message). Fifteen and a half per cent reported receiving phishing messages once or twice every two weeks (frequently receiving suspicious messages). Only a small percentage of respondents (5.5%) said they respond to phishing messages. The vast majority of respondents delete or ignore phishing messages, and some block the number. Table 1 depicts the respondent’s demographic profile.

### 4.2. Measures

A Google form was used to create an online survey with various subscales of self-efficacy, attitude toward behaviour (sharing personal information online), and phishing susceptibility. Each research variable is based on previous research work, with minor changes for contextual consistency.

#### 4.2.1. Self-Efficacy 

This study measured self-efficacy using six survey items adopted from Arachchilage and Love [31]. The scale ranged from 1 = “strongly disagree” to 5 = “strongly agree”.

#### 4.2.2. Attitude towards Sharing Personal Information

Attitude towards sharing personal information online was measured using five survey items adopted from Jansen and van Schaik [30]. A five-point semantic differential scale was adopted as the measurement scale to measure attitude towards behaviour.

#### 4.2.3. Phishing Susceptibility

The current study measured phishing susceptibility using five survey items adapted from Chen et al. [54], with scales ranging from 1 to 7, with 1 indicating “strongly disagree” and 7 indicating “strongly agree.”

### 4.3. Procedures

Several procedures were required to validate the survey instrument used in the study [83]. The procedures began with the development of a survey, followed by an expert review and a pilot test [83]. An expert review was conducted in this study, and the questionnaire was revised based on the expert’s feedback. Following that, a focus group of at least four participants [84] was formed, and a detailed discussion was held to initiate their comments on the survey in this study. The survey questionnaire is shown in Appendix A. A pilot test was conducted based on 54 responses, and all research construct reliability was greater than 0.70 [82].

Finally, some ethical procedures must be followed both before and after gathering information from subjects [85]. This study followed ethical guidelines and was approved by the university Research Ethics Committee. The purpose of the study was fully explained to all participants, and the survey ensured anonymity by not collecting respondents’ personal information. The respondents were fully informed of their other rights, which include confidentiality, privacy, voluntary participation, and the right to withdraw from this study at any time without explanation. 

## 5. Results and Analysis

The SmartPLS 4.0.8.6 version was used as the statistical tool to examine the measurement and structural model as the focus of this paper was to predict the relationships between variables. PLS-SEM, as opposed to covariance-based structural equation modelling (CB-SEM), focuses on predicting how well exogenous constructs predict an endogenous construct [82]. Thus, PLS-SEM, which focuses on the amount of variance explained in the dependent variables, was deemed appropriate for this study.

### 5.1. Normality Assumption

The findings revealed that the data collected were univariate normal. The skewness and kurtosis for all research variables ranged from −0.313 to −0.026 and −0.179 to −0.943, respectively, which met the requirements of univariate normality of the data, which are ±1 and ±7 [86], respectively.

This study examined multivariate skewness and kurtosis as proposed by Cain et al. [87] and Hair et al. [88]. The results showed that the data were multivariate normal, Mardia’s multivariate skewness (β = 1.407, *p* < 0.01), and Mardia’s multivariate kurtosis (β = 14.471, *p* < 0.01). The values of skewness and kurtosis all fall within the criteria of multivariate normality of the data, which are ±3 and ±20, respectively [87].

### 5.2. Common Method Bias

The data collection of the present study was self-reported, and the data for the independent and dependent variables were gathered from the same respondents. There may be an issue of common method bias (CMB). Hence, the statistical procedure needs to be applied in this study to address the CMB issue [89]. Although Harman’s one factor has been commonly applied to detect CMB, literature has noted that it is not appropriate to detect CMB [90]. Researchers argued that “Harman’s test is insensitive, and it is unlikely that a single-factor model will fit the data, especially as the number of variables increases” [91,92,93]. A single-factor result might not be able to explain a significant proportion of the total variance in the dataset, and subsequently, is not able to detect the CMB (potential inflation between variables) [93]. Thus, a full collinearity test was suggested for the detection of the CMB [94].

To test the full collinearity, this study follows the suggestions of Kock [95]. A Variance Inflation Factor (VIF) value of above 3.3 is indicative of potential collinearity problems [95,96]. The results of the current study found that all the VIF values of each research construct ranged from 1.048 to 1.087 which are lower than 3.3, as summarised in Table 2. Hence, the finding indicated that the dataset does not suffer from CMB.

### 5.3. Measurement Model

#### 5.3.1. Validity and Reliability

For the convergent validity, the outer loadings, average variance extracted (AVE), and composite reliability (CR) were assessed. Except for SE6, all of the research constructs’ outer loadings remained within a threshold of 0.708 [97], and most indicators are highly loaded on each construct and significant (−0.490). According to Hair et al. [88], a negative loading item was removed. As a result, SE6 was removed from the research model. As shown in Table 3, all AVE and CR values were greater than 0.50 and 0.70, respectively [86].

#### 5.3.2. Discriminant Validity

The heterotrait–monotrait ratio of correlations (HTMT) method was used to determine discriminant validity [98]. It is proposed that if the HTMT value exceeds 0.85 [98,99], the problem of discriminant validity arises. The findings of the current study in Table 4 indicated that all HTMT values met the suggested criterion of 0.85.

### 5.4. Structural Model

#### 5.4.1. Hypothesis Testing

This study followed the suggestions of Hair et al. [97] by reporting the path coefficients, the standard errors, t-values, *p*-values [100], confidence intervals and effect sizes [101] for the structural model using a 5,000-sample re-sample bootstrapping procedure. Hypothesis H1 of this study posited that there is a negative relationship between self-efficacy and phishing susceptibility (risk of instant messaging phishing victimisation). Despite being significant, self-efficacy had the opposite effect (a positive relationship; β = 0.191, t = 3.336, *p* < 0.001) than what was initially hypothesised (a negative relationship) in the current study. The result, therefore, does not support hypothesis H1.

Hypothesis H2, self-efficacy (β = 0.216, t = 3.916, *p* < 0.001) is positively related to phishing susceptibility (risk of instant messaging phishing victimisation). Thus, H2 is supported. Lastly, attitude towards sharing personal information online (β = −0.147, t = 2.449, *p* < 0.01) was negatively related to and influenced phishing susceptibility (risk of instant messaging phishing victimisation). Thus, hypothesis H3 is supported. Table 5 summarises the results of the hypothesis testing. Figure 2 and Figure 3 depict the research framework’s structural analysis results.

#### 5.4.2. Mediation Analysis

Hypothesis H4 speculates that attitude towards sharing personal information mediates the relationship between self-efficacy and phishing susceptibility (risk of instant messaging phishing victimisation). This study followed Preacher and Hayes´s [102] recommendations to test the mediation of attitude toward sharing personal information online in the relationship between self-efficacy and phishing susceptibility, and indirect effects for mediation were tested using bootstrapping. The indirect effects 95% Boot CI bias-corrected (lower level and upper level) for self-efficacy → phishing susceptibility did not cross zero, indicating there is mediation [102]. Table 6 shows the resulting bootstrapping indirect effects analysis. Results were found statistically significant (β = −0.032, t = 2.077, *p* = 0.038), and as a result, this study supports hypothesis H4. The direct effect (0.191) for the self-efficacy → phishing susceptibility relationship was further assessed, yielding a *p*-value of less than 0.05. The direct effect is still significant, suggesting a complementary partial mediation [103].

#### 5.4.3. Explanatory Power of the Model

The coefficient of determination or R^2^ was examined to determine the explanatory power of the model. As Table 3 shows, the R^2^ value of attitude towards sharing personal information was 0.047 and phishing susceptibility was 0.046. Effect sizes, f^2^ of self-efficacy (0.037) and attitude towards sharing personal information online (0.022) on phishing susceptibility revealed weak effects. In addition, self-efficacy had a small effect (0.049) in producing the R^2^ for phishing susceptibility [104].

#### 5.4.4. Predictive Power of the Model

Shmueli et al. [105] proposed a new evaluation procedure designed specifically for the prediction-oriented nature of PLS-SEM. Therefore, this study expanded the analysis by including a predictive-relevance analysis with PLS-Predict, as suggested by Shmueli et al. [105]. PLSpredict is “a holdout sample-based procedure that generates case-level predictions on an item or construct level” with a 5-fold procedure to check for predictive relevance. Shmueli et al. [105] proposed first checking the latent variable (Q^2^ predict), and if that is greater than zero, then examining the item differences (PLS-LM).

If all of the item differences (PLS-LM) are lower, there is strong predictive power; if all are higher, predictive relevance is not confirmed; if the majority is lower, there is moderate predictive power; and if the minority is lower, there is low predictive power [105]. The Q^2^ for the latent variable phishing susceptibility is 0.011, which is greater than zero, indicating that the construct had a predictive relevance. Following that, based on Table 7, all of the item differences (PLS-LM) were lower than the LM model, confirming that the current research model had a strong predictive power [105].

## 6. Discussion

The overarching goal of this study was to assess the effects of self-efficacy and attitude toward sharing personal information online on the risk of instant messaging phishing victimisation. According to the findings of this study, having the ability to gain anti-phishing knowledge increases the risk of instant messaging phishing victimisation (phishing susceptibility). The results revealed a positive relationship (opposite to the hypothesised direction) between self-efficacy and phishing susceptibility. This study found that having a higher level of self-efficacy (the ability to learn anti-phishing knowledge) increased the risk of being a victim of instant messaging phishing. One plausible explanation is that people try to reduce mental strain by processing information quickly rather than deliberately [30,106]. As a result, because internet users are overconfident in their ability to detect phishing [37], it may be difficult to convince them to take cyber threats seriously [30]. This observation supports the findings of a recent empirical study, which found that self-efficacy was a significant predictor of phishing victimisation risk [62,107]. This finding, however, contradicts previous research that found that self-efficacy had a negligible impact on susceptibility to social engineering attacks [108,109].

In the current study, it was discovered that the belief that one has the ability and resources to acquire anti-phishing knowledge (self-efficacy) has a positive and significant influence on one’s attitude toward sharing personal information online. This finding supports the findings of several studies that self-efficacy is significantly related to attitude formation [110,111]. The previous study’s finding of an insignificant relationship between self-efficacy and attitude formation, on the other hand, was contradicted [112,113].

According to the current study, a negative attitude toward sharing personal information online significantly predicted susceptibility to phishing (risk of instant messaging phishing victimisation). This finding corroborates previous research that found that attitude toward behaviour influenced the risk of cyber-enabled crime (burnout and violent victimisation) [72] and cyber-dependent crime (phishing victimisation) [30,32]. This observation, however, contradicts Espelage et al.’s [114] research, which indicates that regardless of attitude toward risky behaviour, it does not necessarily protect individuals from being victimised. More specifically, this study demonstrates that a person who has a negative attitude toward sharing information online is more likely to want to protect his or her information [115], decreasing the risk of becoming a phishing victim. This is due to the fact that if an internet user is not confident that his or her personal information will be handled appropriately and carefully, he or she may develop a more unfavourable (negative) attitude toward sharing information online [116].

The mediation results demonstrated that attitudes toward sharing personal information online did play a role in predicting the relationship between self-efficacy and phishing susceptibility. In other words, this study suggests that a negative attitude towards sharing personal information online mediates the relationship between one’s ability to gain anti-phishing knowledge (self-efficacy) and phishing susceptibility. When a negative attitude toward sharing personal information online intervenes, the strength of this relationship deteriorates. One plausible explanation for this finding is that detecting online fraud necessitates extensive knowledge of the fraud [117]. Scholars have suggested that a person’s attitude toward resolving fraud or spam issues can be callous [118]. This is due to the fact that the vast majority of internet online fraud is expected to spread quickly [119], making it difficult for individuals to combat it effectively [118].

### 6.1. Theoretical Implications

PMT identifies several predictors that lead to the intention to implement the recommended precautionary measures [47]. According to researchers [120,121], the use of PMT in the domain of phishing is extremely limited. Jansen and van Schaik [122] found that PMT can be directly applied to the domain of phishing, where self-efficacy increases self-reported precautionary behaviour when securing information and sharing online information. According to the PMT theory, one’s attitude change is influenced by one’s protection motivation behaviour [47]. People with high self-efficacy, for example, are more likely to change their attitude, allowing them to make better decisions [6]. Because of the nature of PMT in examining human protection motivation behaviour, an increasing number of studies have applied it to phishing [30,32,55]. This is because when a person has a positive attitude toward online precautionary behaviour (i.e., a negative attitude toward online information sharing) as well as a high perceived self-efficacy, he or she believes that phishing attackers will not compromise him or her [6,30]. As a result, the purpose of this study is to close the research gap by confirming that self-efficacy and attitude as PMT measures significantly predict the risk of phishing victimisation, particularly the risk of instant messaging phishing victimisation.

There was a contradictory study result on the significance of self-efficacy as a predictor of attitude towards protection behaviour [32]. The non-significant result seems strange and surprising as most studies indicated that self-efficacy was the strongest predictor of attitude towards protection behaviour [51,52,67]. As a result, Martens et al. [32] requested that the relationship between self-efficacy and attitude toward protective behaviour be investigated. Because of the non-significant result, the researchers concluded that self-efficacy was losing explanatory power in predicting attitudes toward behaviour in the phishing context [32,123]. As a result, this study has made a theoretical contribution to the understanding of a significant relationship between self-efficacy and attitude formation.

Attitude served as a mediator in predicting the offender’s behaviour in both offline and online contexts [124,125]. On the other hand, an individual’s (victim’s) attitude was also relevant in examining the context of online security behaviour [30]. In the context of intent to engage in precautionary online behaviour, the attitude was defined as attitude toward personal information-sharing online [30]. Precautionary measures help to protect internet users from phishing attacks [30,32]. Thus, using PMT as a model, this study used attitude towards behaviour as both a predictor and a mediator to predict the risk of instant messaging phishing victimisation. Without denying that attitude was an important factor in explaining offender behaviour, the findings of this study contribute to the body of knowledge in this field by indicating that attitude is an important factor in predicting victim behaviour.

### 6.2. Practical Implications

The current study discovered that a higher level of self-efficacy increases one’s risk of being a victim of instant messaging phishing. This study also discovered that having a higher level of self-efficacy reduces one’s susceptibility to instant messaging phishing victimisation if one has a negative attitude towards sharing personal information online. Since this study found that self-efficacy (or confidence in one’s ability to acquire anti-phishing knowledge) can lower the likelihood of being a victim of instant messaging phishing, it does show that the ability of Malaysian internet users to acquire anti-phishing knowledge is not a problem. The current study’s findings emphasise the importance of self-efficacy as the underlying principle for implementing security behaviour. The ability of internet users to practise security practises when unsupervised is critical. Therefore, it is suggested that phishing-related awareness, education, and training programmes could be continued to increase self-efficacy [63,126]. Every internet user may begin with a different level of technical knowledge, competence, and awareness; thus, those who run anti-phishing efforts and informational websites may need to have empathy for and an understanding of this fact. The level of cybercrime awareness may be stated and quantified, for example, by the inclusion of quiz gaming that enables internet users to show their comprehension of phishing.

According to the literature, privacy expectations are decreasing as people become more comfortable disclosing personal information [127]. The internet’s popularity has enabled internet users to engage and share significant personal information or experiences on online platforms [128,129]. According to existing literature, people believe that sharing personal information can result in benefits rather than privacy risks [130]. People tend to ignore the risks of disclosing information (high risk-taking attitudes) because they believe they are immune to cyber threats [130,131]. Therefore, it is understandable that guiding internet users to have a negative attitude toward sharing personal information online can be a challenging task. It is suggested that government agencies responsible for combating phishing attacks inform and remind internet users on a regular basis to make informed sharing decisions in order to foster a negative attitude toward sharing personal information online. For example, in the anti-phishing awareness campaign, internet users are advised to use information verification before disclosing personal information online.

According to the findings of this study, one’s attitude toward sharing personal information online can be influenced by one’s level of self-efficacy. It does demonstrate that a high level of ability to acquire anti-phishing knowledge, which assists internet users in acquiring anti-phishing knowledge independently, can lead to a negative attitude toward sharing personal information online. When a person is familiar with anti-phishing techniques, he or she has a strong perception of privacy risk. As a result, he or she may regard information sharing as risky behaviour and, as a result, have a negative attitude toward sharing personal information online [132,133]. As a result, it is recommended that internet users regularly update their phishing-related knowledge, such as learning more about privacy risks, reporting any unknown or suspicious messages, and blocking or restricting any unknown senders on instant messaging applications. This is due to heightened awareness of the risks associated with disclosing personal information, which results in limited disclosure and information protection [134], as well as a negative attitude toward sharing personal information online [115].

## 7. Limitations of the Study and Validity Threats

One of the limitations of this study is that the survey’s respondent age limit of up to 43 years old may limit the findings’ applicability to all Malaysians. Because the current study’s respondents are educated and experienced internet users, the study’s findings may not apply to other populations who are less educated or technologically savvy. Future research may replicate the current study’s research framework to investigate phishing susceptibility (risk of instant messaging phishing victimisation) among Malaysians of various ages.

Threats to External Validity [135,136] compromise the generalisability of the findings [20]. Although this study cannot claim that the empirical evaluation’s findings are generalizable, it is expected that they will not change much by analysing the findings using a longitudinal study. This is because the current study used a purposive sample technique for data collection, with filtering questions (refer to Table A1) used to ensure respondents met the screening requirements and, as a result, establish the validity of the results. Furthermore, follow-up longitudinal studies (e.g., interviews and focus groups) might be done to validate the study’s external validity.

Threats to Internal and Construct Validity [135,136]: Internal validity is concerned with the researcher’s interpretation and bias of the data, whereas construct validity is concerned with the potential risks/threats associated with the study design [20]. To reduce the threats to construct validity, the survey questionnaires were drawn from the existing research literature (to reduce bias in design or any effect on the results or interpretation). Furthermore, an expert review was done to ensure the validity of the survey (refer to Section 4.3).

## 8. Conclusions

As cited in Martens et al.´s [32] research, an increasing number of studies are being conducted to investigate cybercrime using variations of the protection motivation theory (PMT) [137,138]. In addition to self-efficacy and attitude toward behaviour (protection motivation), PMT includes several other variables, including perceived severity and response efficacy [32]. However, only self-efficacy and attitude were used as predictors of the risk of instant messaging phishing victimisation in this study. This is because the current study’s main goal was to examine the factors of phishing susceptibility rather than measuring one’s perception of the consequences of being victimised and not being victimised [30]. As a result, the current study excluded perceived severity and response efficacy. A future study could build on the current research framework by incorporating these two variables as moderators and examining whether the perceptions of the consequences differed between high and low self-efficacy internet users. Finally, the findings may point to major variations in cyber-security posture based on gender, age group, etc. [19,139]. Future possibilities include performing a strata analysis while taking into account numerous additional demographic factors that can be statistically tested.

## Figures and Tables

**Figure 1 ijerph-20-03514-f001:**
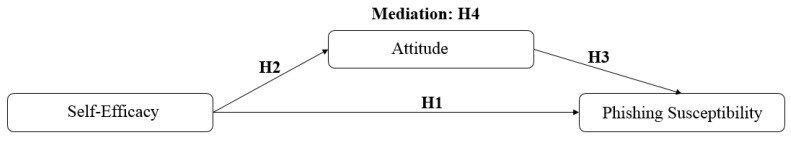
Research Framework.

**Figure 2 ijerph-20-03514-f002:**
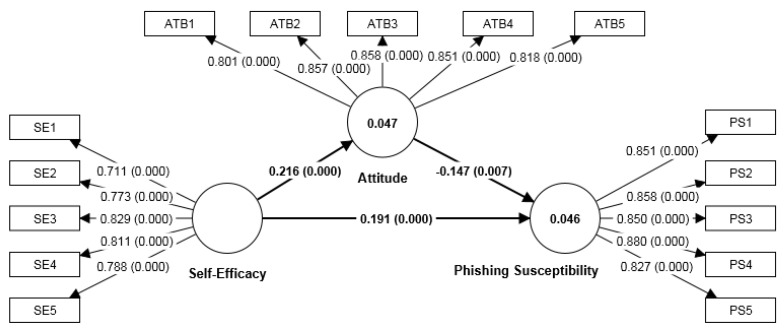
Structure model. Note: Inner model (paths) indicates the coefficients and *p*-values in parenthesis. Outer model (paths) indicates the item loadings and *p*-values in parentheses. Constructs ATB and PS show the R^2^ value.

**Figure 3 ijerph-20-03514-f003:**
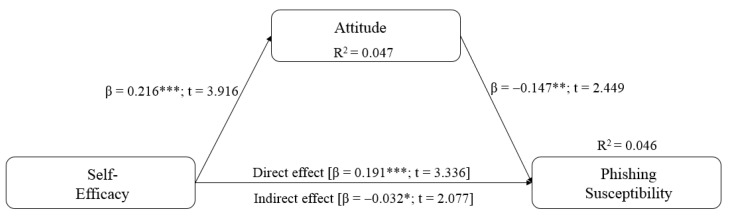
Research framework’s structural analysis results. Note: *** = *p* < 0.001, ** = *p* < 0.01, * = *p* < 0.05.

**Table 1 ijerph-20-03514-t001:** Demographic Profile.

Respondent Characteristics (N = 328)	Frequency	Per cent
Gender		
Male	151	46
Female	177	54
Age (Gen-Z)		
18 to 22 years old	133	40.5
23 to 27 years old	195	59.5
Current Location		
Northern Region (Kedah, Perak, Perlis, Pulau Pinang)	47	14.3
Central Malaysia (Federal Territory: Kuala Lumpur, Putrajaya, Labuan), Negeri Sembilan, Selangor	155	47.3
Southern Region (Johor, Melaka)	78	23.8
East Coast (Kelantan, Pahang, Terengganu)	12	3.7
East Malaysia (Sabah, Sarawak)	36	11.0
Educational Level		
PhD	6	1.8
Master’s Degree	49	14.9
Bachelor’s Degree	239	72.9
Diploma	23	7.0
Technical/Vocational Education & Training	4	1.2
Secondary School	6	1.8
Others (Foundation)	1	0.3

**Table 2 ijerph-20-03514-t002:** Full Collinearity Testing.

Attitude	Phishing Susceptibility	Self-Efficacy
1.071	1.048	1.087

**Table 3 ijerph-20-03514-t003:** Convergent validity.

	Outer Loading	Composite Reliability	Average Variance Extracted (AVE)	R^2^
Attitude (ATB)		0.921	0.701	0.047
The sharing of personal information online is.				
ATB1	0.801			
ATB2	0.857			
ATB3	0.858			
ATB4	0.851			
ATB5	0.818			
Phishing Susceptibility (PS)		0.930	0.728	0.046
… becoming/become victimised by instant messaging phishing attacks.				
PS1	0.851			
PS2	0.858			
PS3	0.850			
PS4	0.880			
PS5	0.827			
Self-Efficacy (SE)		0.888	0.614	
I could successfully gain anti-phishing knowledge if…				
SE1	0.711			
SE2	0.773			
SE3	0.829			
SE4	0.811			
SE5	0.788			
* SE6	-			

Notes: * item removed due to lower loading.

**Table 4 ijerph-20-03514-t004:** Heterotrait–Monotrait Ratio of Correlations (HTMT).

	Attitude	Phishing Susceptibility	Self-Efficacy
Attitude			
Phishing Susceptibility	0.112		
Self-Efficacy	0.232	0.175	

**Table 5 ijerph-20-03514-t005:** Hypothesis Testing.

Hypothesis		Std. Beta (β)	Std. Error	t-Value	BCI LL	BCI UL	*p*-Value	Results	f^2^	Effect Size	VIF
					5%	95%					
H1	SE → PS	0.191	0.057	3.336	0.081	0.270	<0.001	Not Supported	0.037	Small	1.049
H2	SE → ATB	0.216	0.055	3.916	0.109	0.292	<0.001	Supported	0.049	Small	1.000
H3	ATB → PS	−0.147	0.060	2.449	−0.237	−0.038	0.007	Supported	0.022	Small	1.049

Note: SE = Self-efficacy, ATB = Attitude towards sharing personal information online, PS = Phishing Susceptibility, BCI = Confidence Interval Bias Corrected, UL = upper level, LL = Lower level.

**Table 6 ijerph-20-03514-t006:** Mediation Testing.

Hypothesis		Indirect Effect							Direct Effect				Mediation
		Std. Beta (β)	Std. Error	t-Value	*p*-Value	BCI LL	BCI UL	Results	Std. Beta (β)	Std. Error	t-Value	*p*-Value	
						2.5%	97.5%						
H4	SE → ATB → PS	−0.032	0.015	2.077	0.038	−0.062	−0.003	Supported	0.191	0.057	3.336	<0.001	Complementary Partial Mediation

Note: SE = Self-efficacy, ATB = Attitude Towards Sharing Personal Information Online, PS = Phishing Susceptibility, BCI = Confidence Interval Bias Corrected, UL = Upper Level, LL = Lower Level.

**Table 7 ijerph-20-03514-t007:** PLS-Predict Summary.

Construct	Q² Predict			
Phishing Susceptibility (PS)	0.011			
Items	PLS	LM	PLS-LM	Q² Predict
	RMSE	RMSE	RMSE	
PS1	1.681	1.704	−0.023	−0.002
PS2	1.627	1.654	−0.027	0.012
PS3	1.536	1.548	−0.012	0.001
PS4	1.494	1.517	−0.023	0.022
PS5	1.573	1.589	−0.016	0.013

Notes: RMSE = root mean squared error; PLS = partial least squares path model; LM = linear regression model; Q^2^ predict = predictive relevancy.

## Data Availability

Data supporting reported results are available from the authors.

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
