# Peer review of "Thwarting Instant Messaging Phishing Attacks: The Role of Self-Efficacy and the Mediating Effect of Attitude towards Online Sharing of Personal Information"

_ijerph, 2023, doi:10.3390/ijerph20043514_

Round 1

Reviewer 1 Report

The authors investigate the effects of self-efficacy and attitude toward sharing personal information online on the risk of instant messaging phishing attacks. The study was conducted using data from 328 active Malaysian instant messaging users using partial least squares structural equation modeling. They found that a higher level of self-efficacy and a negative attitude toward sharing personal information online were significant predictors of phishing susceptibility.

My comments on your work are:

1) It is recommended to include precise statistics data of the problem studied to make its severity clear, specifically phishing attacks and their consequences.

2) Indicate whether the data was collected from users across the country or from specific locations.

3) “This study received 353 questionnaires in total. Following a thorough examination of the 335 datasets, a total of six datasets.” (lines 262 and 263). Numbers do not match.

4) It is suggested to present the data of the participants in tables.

5) It is suggested that an analysis by stratum be carried out in future work. That is, age groups, gender, platform, and so on.

6) Figures need to be improved.

Author Response

Dear Reviewer, 

We sincerely appreciate all valuable comments and suggestions, which helped improve the manuscript's quality. We have addressed all comments to the best of our ability. The attachment contains a complete list of all the changes. Please see the attachment labelled "Author to respond reviewer 1.

Thank You.

Reviewer 2 Report

Dear author and reviewers. The research is very complete and well structured. I liked it very much and I see it very well structured. Only, I think that the introduction could be improved by adding some meta-analysis study.

Author Response

Dear Reviewer, 

We sincerely appreciate all valuable comments and suggestions, which helped improve the manuscript's quality. We have addressed all comments to the best of our ability. The attachment contains a complete list of all the changes. Please see the attachment labelled "Author to respond reviewer 2.

Thank You.

Reviewer 3 Report

1. In the introduction alone, 36 references were used. I would suggest citing one or a maximum of two references at a time.

2. Typo - cit-izens

3. Validity threats should be included for better readability of the study. (Validity threats of the study can be seen in the study by Rubia Fatima et al. "How persuasive is a phishing email? A phishing game for phishing awareness"

 4. I would suggest writing a structured abstract for better readability. Such as: Context, Objective, Method, Results, and Conclusion.

5. Add the flow of the paper at the end of the introduction. For example, Section 2 explains..., Section 3 discusses... etc

6. How do you view the different methods of counteracting attacks? Is there a discussion missing about your point of view? For more information, see Rubia Fatima et al. "How persuasive is a phishing email? A phishing game for phishing awareness" ).

7. In the study, one of the aspects was "Attitude towards sharing personal information..." This was an intriguing aspect you might have missed. Would you mind letting me know if the study "Sharing information online rationally: An observation of user privacy concerns and awareness using serious games" is relevant and can provide some more perspective?

 8. Be sure to clearly mention the research gap in the study. We can see many similar studies published online. (one of the differences is that your study is focused on Malaysian participants?)

9. Add a survey questionnaire at the end of the study, as well.

Author Response

Dear Reviewer, 

We sincerely appreciate all valuable comments and suggestions, which helped improve the manuscript's quality. We have addressed all comments to the best of our ability. The attachment contains a complete list of all the changes. Please see the attachment labelled "Author to respond reviewer 3.

Thank You.

Round 2

Reviewer 3 Report

The authors have answered and responded to all the raised concerns.